# Ossification and Fusion of the Vertebral Ring Apophysis as an Important Part of Spinal Maturation

**DOI:** 10.3390/jcm10153217

**Published:** 2021-07-21

**Authors:** Lorenzo Costa, Steven de Reuver, Luc Kan, Peter Seevinck, Moyo C. Kruyt, Tom P. C. Schlosser, René M. Castelein

**Affiliations:** Department of Orthopaedic Surgery, University Medical Center Utrecht, 3584CX Utrecht, The Netherlands; l.costa-2@umcutrecht.nl (L.C.); S.deReuver-4@umcutrecht.nl (S.d.R.); l.p.m.vankan@students.uu.nl (L.K.); p.seevinck@umcutrecht.nl (P.S.); M.C.Kruyt@umcutrecht.nl (M.C.K.); T.P.C.Schlosser@umcutrecht.nl (T.P.C.S.)

**Keywords:** ring apophysis, maturation, ossification, fusion, scoliosis

## Abstract

In scoliosis, most of the deformity is in the disc and occurs during the period of rapid growth. The ring apophyses form the insertion of the disc into the vertebral body, they then ossify and fuse to the vertebrae during that same crucial period. Although this must have important implications for the mechanical properties of the spine, relatively little is known of how this process takes place. This study describes the maturation pattern of the ring apophyses in the thoracic and lumbar spine during normal growth. High-resolution CT scans of the spine for indications not related to this study were included. Ossification and fusion of each ring apophysis from T1 to the sacrum was classified on midsagittal and midcoronal images (4 points per ring) by two observers. The ring apophysis maturation (RAM) was compared between different ages, sexes, and spinal levels. The RAM strongly correlated with age (R = 0.892, *p* < 0.001). Maturation differed in different regions of the spine and between sexes. High thoracic and low lumbar levels fused earlier in both groups, but, around the peak of the growth spurt, in girls the mid-thoracic levels were less mature than in boys, which may have implications for the development of scoliosis.

## 1. Introduction

The majority of pediatric spinal deformities develop during puberty when the body weight and dimensions increase rapidly, as the skeleton matures into its adult form.

Recent studies have demonstrated the significant contribution of the intervertebral discs to the deformation of the spine in scoliosis [1,2,3]. For undisturbed and harmonious spinal development, rapidly increasing loads during the growth spurt require adequate maturation of the spine’s stabilizers, of which the disc is an essential component [4]. Skeletal maturity is traditionally assessed on X-rays of the iliac crest, the hand, and wrists, or other growth cartilages remote from the spine. The most used classifications are the Risser grade based on ossification and fusion of the iliac apophysis and the Sanders simplified skeletal maturity scoring system, based on the maturation of the hand epiphyses [5,6]. Another classification is the Proximal Humerus Ossification System (PHOS) [7]. This classification is a five-stage system that uses the proximal humeral physis in assessing skeletal maturity. These skeletal maturation scoring systems correlate with the Peak Height Velocity (PHV) during pubertal growth and the curve acceleration phase in scoliosis, however, they do not necessarily assess the maturation of different regions of the spine itself [6,8,9,10].

After the formation of the three primary ossification centers for each vertebra (one for the body and two for the vertebral arch), the maturation of the spine continues with the closure of the neurocentral synchondroses from age 4–8 [11]. The secondary ossification centers are the ring apophyses and the tips of the transverse and spinous processes [12]. The ring apophyses are not inside the epiphyseal plate and are not involved in the longitudinal growth of the spine [13]. During growth, they encircle the inferior and superior surfaces of the vertebral bodies. The outermost fibers of the annulus fibrosus, the Sharpey’s fibers, insert into the ring apophysis and thus anchor the intervertebral disc to the two adjacent vertebrae (Figure 1) [14,15,16,17,18]. During skeletal maturation, these initially cartilaginous insertions of the disc ossify and ultimately fuse to the vertebral bodies [19]. This process has important implications for the mechanical stability of the disc-vertebral body complex at a time when spinal loading increases rapidly, but very little is known about its different phases during growth [20,21].

## 2. Materials and Methods

### 2.1. Study Population

After a waiver from the ethical review board (ERB) for formal review of this retrospective study, pre-existing high-resolution Computed Tomography (CT) scans of the thorax and/or abdomen acquired from a tertiary pediatric hospital for indications not related to this study (e.g., trauma screening, pulmonary disease, and gastro-enteric disorders) were included from our local patient archiving and communications system (PACS). Inclusion criteria were patients between 6 and 21 years of age and available high-resolution images of the spine (slice thickness ≤ 1 mm and slice interval ≤ 1.5 mm). The ages of 6 and 21 were chosen based on previous studies by Woo et al. and Uys et al. [13,19]. Exclusion criteria were the presence of spinal pathology, bone disorders (e.g., Scheuermann’s disease), syndromes associated with growth disorders, growth hormone treatment, and insufficient CT-scan quality including movement artifacts. According to all available information, subjects represented healthy adolescents. A minimum of 10 subjects per age cohort was included. Age, sex, Risser grade, and proximal humeral ossification system (PHOS) were collected on the coronal survey scans. 

### 2.2. CT-Scan Analysis

The included CT scans were analyzed independently by two trained observers, who scored each vertebra separately blinded from each other. Multiplanar images of the exact mid-sagittal and mid-coronal plane of each individual vertebra were reconstructed, using the RadiAnt DICOM viewer© (RadiAnt, Poznan, Poland) (Figure 2). 

The window level was set to the bone. Ossification and/or fusion of the anterior, posterior, and lateral parts of each superior and inferior apophyseal ring was classified. If needed, the two observers viewed adjacent slices to confirm that suspected ossified structures were part of the ring apophysis, and discrepancies were resolved by consensus.

According to previous observations by Uys et al., and confirmed following a pilot study performed, the ossification and fusion of each region of interest (ROI) of the ring apophyses were scored as shown in Figure 3 [19]:Phase 0: Ring not detectable.Phase 1: Ring detectable.Phase 2: Fusion not completed.Phase 3: Fusion completed.

Next, the overall ring apophysis maturation (RAM) of each ring was classified as:Stage 0: no ossification (phase 0) in all 4 ROI.Stage 1: Beginning of ossification (phase 1 in 1–3 ROI).Stage 2: Complete ossification (phase 1 in all 4 ROI).Stage 3: Incomplete fusion (phase 3 in 1–3 ROI).Stage 4: Complete fusion at all 4 points (phase 3 in all 4 ROI).

Intraclass correlation coefficients (kappa value) were calculated for the assessment of intra- and inter-observer reliability.

### 2.3. Statistical Analysis

Statistical analyses were performed using SPSS 25.0 for Windows (IBM, Armonk, NY, USA). The median, range, and IQR of the RAM were calculated for each age and spinal level for both sexes. The normality of distribution of the RAM within the study population was analyzed via Q-Q plots. The correlation between the RAM and age was tested with a non-parametric Spearman’s rank test as well as for different areas of the rings. In the ring ossification stage (phase 1), the authors compared the sagittal plane (both anterior and posterior) with the coronal plane, with a standard *t*-test. The same procedure was done for fusion (phase 3). Different growth patterns between the lower thoracic and thoracolumbar spine and the other spinal sections were calculated through a generalized linear model. The correlation between the median RAM of the whole spine and the conventional skeletal maturity scores (Risser and PHOS) was analyzed with a Spearman’s rank test. The *p*-value was set at 0.05. 

## 3. Results

### 3.1. Study Population 

Out of 4775 available CT scans, 456 could be included in this study. Most exclusions were due to insufficient image quality. Of the included subjects, 50% were females. The CT scan images analyzed were 289 (63%) full-body, 82 (18%) thoracic, and 85 (19%) abdominal. Descriptive statistics of patients and CT scans are shown in Table 1.

### 3.2. Ring Apophysis Maturation

Maturation of the ring apophysis is a process that varies for each age at different levels of the spine and differs between sexes. Furthermore, it does not strictly follow the same patterns and timing of the most common physes used for skeletal maturity assessment such as iliac apophysis (Risser), hand epiphysis (Sanders), and proximal humeral physis (PHOS).

Ossification occurred from age 9–15 in males and 7–15 in females, fusion from 14–19 and 13–19, respectively. 

RAM correlated significantly with age (R = 0.892, Figure 4) and both ossification and fusion occurred earlier in females (*p* = 0.002, Figure 4). 

At age 21, 98% of the rings were completely fused (stage 4). Whereas ossification and fusion occurred on average between 9 and 19 in males and 7 and 19 in females, important differences were observed per spinal level, especially when related to the average age of the PHV (13 years of age in females and 15 in males) as can be seen in Table 2 and Figure 5. Most differences between the sexes could be detected in the thoracic levels. In females at the age of 13, the median of RAM was stage 0 between T1 and T6 and stage 1 between T7 and T12. In 15-year old males, the median of maturation in spinal sections was stage 3 for T1 and T2 and stage 1 between T3 and T12. 

For both males and females, the ossification of the inferior ring was earlier than the superior ring. In contrast, fusion occurred earlier in the superior ring. Furthermore, maturation of one ring does not appear to occur in all areas at the same time: ossification (phase 1) and fusion (phase 3) occurred half a year later (between 11 and 12 years, median 12) in the coronal than the sagittal plane (*p* = 0.031). Finally, the high thoracic and low lumbar levels ossified later but fused earlier (while the growth spurt was ceasing in females and was mid-way in males) than the thoracolumbar levels (after the growth spurt has ceased in females and was ceasing in males).

### 3.3. Correlation with Other Skeletal Maturity Parameters

Although ring maturation varies per level studied, the overall RAM presented a clear correlation with the other two classifications. The Spearman’s test between the RAM and Risser grade was 0.900 and between RAM and PHOS was 0.908.

## 4. Discussion

This study provides a CT-based analysis of the maturation of the disc’s fixation to the vertebral body, the ring apophysis, related to age, sex, and spinal levels. Knowledge of the maturation pattern of the ring apophysis is important in the management and etiologic understanding of developmental spine problems since the disc is considered the primary passive stabilizer of the spine [4,23,24]. Whether it is anchored to bone or cartilage during a period in life when body weight and dimensions increase rapidly is supposed to make a major difference for the mechanical properties of the system.

Unlike what all traditional maturation parameters that Risser, PHOS, and Sanders suggest, spinal maturation differs per spinal level. Overall, ossification occurred from age 9–15 years in males and 7–15 in females, fusion from 14–19 and 13–19. Between 12–15 in females and 13–16 years old in males, the ring apophysis undergoes a massive change [25,26]. In girls, around their growth spurt, fewer vertebrae have started the maturation process than in boys, who, in general, have a growth spurt around two years later (Table 2) [22]. 

The anterior and posterior parts, compared to the lateral parts, ossified and fused half a year earlier and the high thoracic and low lumbar levels fused earlier than the mid-thoracic and thoracolumbar.

We observed an earlier fusion of the superior ring compared to the inferior ring as was also observed in earlier radiography studies [16,19]. Moreover, the observations that the inferior ring ossifies earlier than the superior one, that they have the same level of maturation at the age of 15–16, and finally that the superior ring overcomes the inferior one during the fusion stage is in line with the findings by Woo et al. [13].

Not all of the spinal areas mature at the same time. This study showed a later fusion in the mid thoracic and thoracolumbar spine in both sexes, which is similar to the closure pattern of the Neuro Central Junction [11]. Interestingly, the most common curve type in Adolescent Idiopathic Scoliosis (AIS) patients is in the same region where the maturation of the ring apophysis is slower [27]. 

As mentioned previously, understanding the maturation of the spine is of key importance for the management of disturbances of its harmonious development. Nowadays, the most used technique to determine bone maturation is the Risser grade even though its accuracy is debated since the Risser stages do not reflect the exact growth activity in the vertebral endplates [6,9,10,28,29]. Even though other classifications such as Sanders are shown to be more reliable, a specific, spine-based classification of spine maturation is lacking. Furthermore, the spine continues to mature after Risser 4 and 5. Similar discrepancies have already been demonstrated by James et al. in 1958 in which in most cases of the studied x-rays, Risser 5 was not synchronous with the end of the ring maturation [18]. This delayed maturation, as compared to most of the long bones, may be relevant to better understand the response of the spine to the increased loads of the adolescent body during the growth spurt. This topic is nicely displayed in the paper by Sanders et al. (2020) [30]. The authors explained that the spine continues to grow longer than the lower extremities [30]. Di Meglio et al. provided similar results in two of their studies, showing differences in yearly height gain velocity between the trunk and the lower limbs [22,31]. Furthermore, as the pelvis reflects the lower extremities more, it is clear that the Risser grade is not deeply connected to the growth of the spine which continues after the lower limb’s growth has ceased [30]. Moreover, it is clear that Risser 1 occurs after the peak height velocity as shown previously by Di Meglio et al. and in Figure 5 while ring apophysis maturation varies, depending on spinal level, for each Risser stage [22]. 

Finally, inter- and intra-observer reliability of RAM was highly positive, resulting in a substantial agreement. Nevertheless, many differences between the RAM and other classifications could be detected. This might be due to the necessity to form age groups of the subjects.

This study gives an important insight into the maturation of the spine itself, showing interesting differences between the sexes and different anatomical areas. The area of the spine in which most common types of idiopathic scoliosis develop appears to mature later than the rest of the spine. Sharpey’s fibers insert into the ring apophysis and thus anchor the intervertebral disc to the two adjacent vertebrae [19]. As it is a weak point, this ossification and fusion process might have important implications for the mechanical stability of the disc–vertebral body complex of the mid-thoracic and thoraco-lumbar sections at a time when spinal loading increases rapidly due to the growth spurt [20,21]. This may be important for understanding the patho-mechanism of idiopathic scoliosis. Furthermore, around the PHV females appear to have less matured rings if compared to males at the PHV. As most adolescent spine deformities occur mainly in this period and females have a less-matured spine, this could partly explain why the onset of these deformities is more common in females. 

This study used an existing CT database to analyze ossification and fusion of the ring apophysis, obviously, this cannot be used in clinical practice because of radiation hygiene [32]. We are presently working on the further development of bone-MRIs, a new radiation-free technique, which uses MRI to create synthetic CT images based on deep-learning processes [33,34]. Possibly, in the near future, it can also be applied to the scoliotic spine, providing a true spine-based assessment of spinal maturation per level of the spine in a patient group that often undergoes MRI scanning as a regular procedure.

## 5. Conclusions

This study describes the maturation of the ring apophysis as the attachment of the disc to the vertebral body on CT images and shows that they ossify and fuse later in the mid-thoracic and thoraco-lumbar spine. Furthermore, related to the timing of their growth spurt, the female spine appears to be less mature than the male spine.

## Figures and Tables

**Figure 1 jcm-10-03217-f001:**
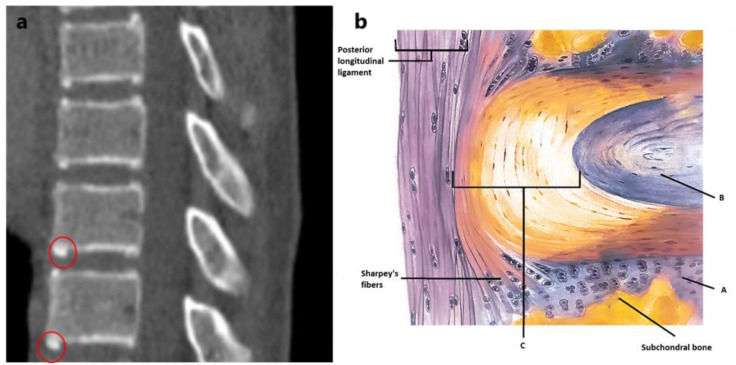
(**a**) A view of the ossified rings and the start of the fusion (red circles) to the vertebral body; (**b**) schematic anatomical view of the IVD and the attachment of the Sharpey’s fibers to cartilage tissue. A: cartilaginous endplate. B: Nucleus pulposus. C: Anulus fibrosus.

**Figure 2 jcm-10-03217-f002:**
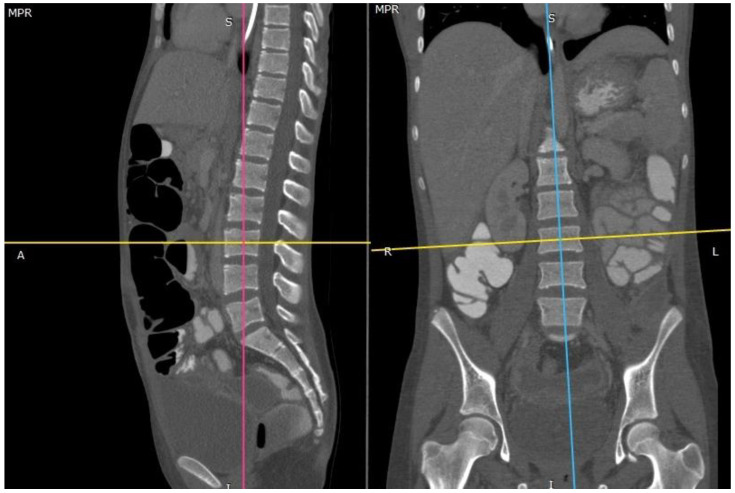
Mid-sagittal and mid-coronal reconstructions were used for each spinal level to describe the presence of ossification and fusion in four areas of each ring.

**Figure 3 jcm-10-03217-f003:**
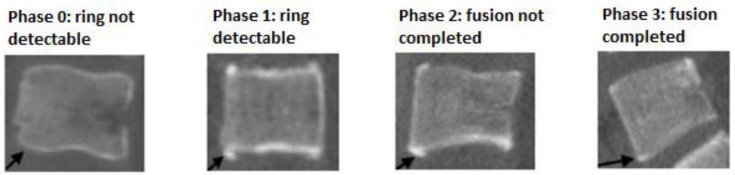
The four phases of maturation of the ring apophysis on mid-sagittal images. In phase 0 the rings are still in a cartilaginous stage and are not detectable on CT scans. In phase 1 the rings are ossified and can be seen on CT scans but have not yet started to fuse. In phase 2 the rings are starting to fuse with the bodies. In phase 3 the rings are completely fused with the vertebral bodies.

**Figure 4 jcm-10-03217-f004:**
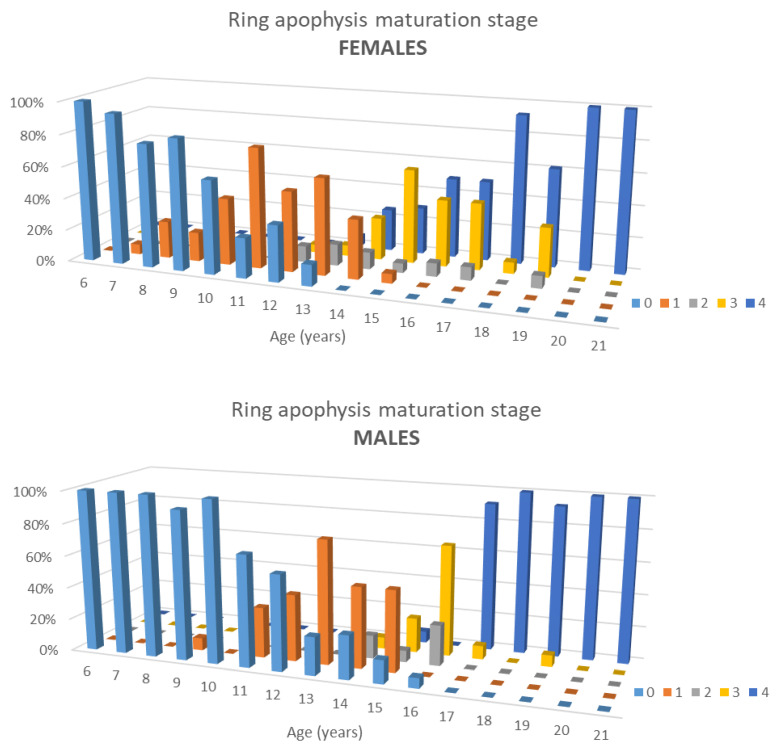
A 3-D histogram showing the percentiles of the different maturation stages of the apophyseal ring in females and males at different ages.

**Figure 5 jcm-10-03217-f005:**
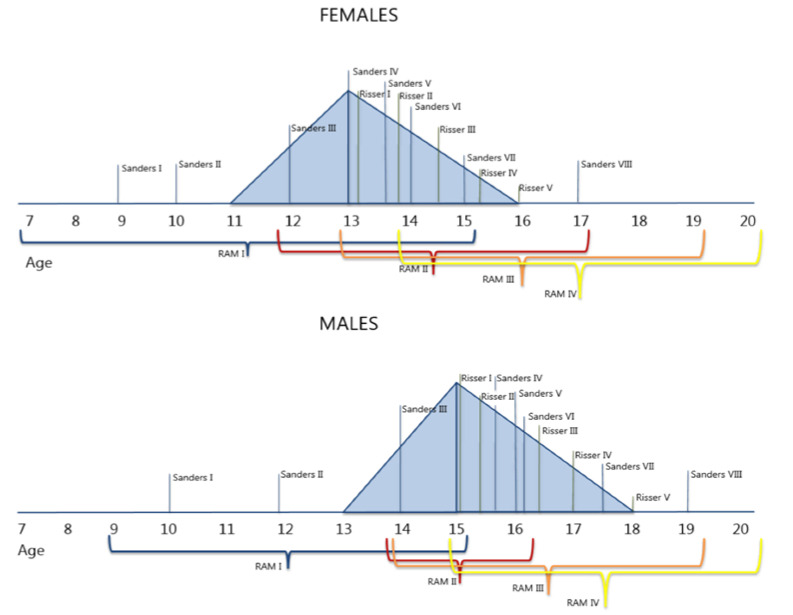
Figure based on Di Meglio et al. (2011) in which Sanders, Risser, and RAM classifications are correlated to growth velocity (and peak height velocity) in females and males [22].

**Table 1 jcm-10-03217-t001:** Patient demographics.

Population		
Age	Males	Females
6	15	17
7	18	16
8	13	13
9	13	11
10	11	12
11	16	16
12	17	20
13	17	15
14	14	19
15	14	17
16	16	12
17	14	13
18	12	14
19	14	13
20	12	11
21	11	10
Total (percentage)	227 (49.7%)	229 (50.3%)
Mean age	13.22	13.15
SD	4.52	4.44
Range	6–21	6–21
CT scans		
Selected CT scans	456	
Total-body n (%)	289 (63%)	
Thoracic n (%)	82 (18%)	
Abdominal n (%)	85 (19%)	

**Table 2 jcm-10-03217-t002:** The differences in mean maturation of the apophyseal ring in males (in blue) and females (in red) for each spinal level.

							M	0	1	2	3	4							
							F	0	1	2	3	4							
AGE	10	11	12	13	14	15	16	17	18	AGE	10	11	12	13	14	15	16	17	18
**T1**										**T1**									
**T2**										**T2**									
**T3**										**T3**									
**T4**										**T4**									
**T5**										**T5**									
**T6**										**T6**									
**T7**										**T7**									
**T8**										**T8**									
**T9**										**T9**									
**T10**										**T10**									
**T11**										**T11**									
**T12**										**T12**									
**L1**										**L1**									
**L2**										**L2**									
**L3**										**L3**									
**L4**										**L4**									
**L5**										**L5**									
**S1**										**S1**									

## Data Availability

The data presented in this study are available on request from the corresponding author. The data are not publicly available due to ethical and privacy reasons.

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
