# Peer review of "Ossification and Fusion of the Vertebral Ring Apophysis as an Important Part of Spinal Maturation"

_jcm, 2021, doi:10.3390/jcm10153217_

Round 1

Reviewer 1 Report

This is an interesting paper nicely showing the processes of spinal maturity. The authors have done an excellent job shedding light on this important topic, I do however have some comments and questions.

1. Were the reasons and conditions for CT-scanning for the subjects of such nature that it could disrupt the skeletal maturity process? In other words, are these subjects representable for healthy adolescents?

2. A significant number of individuals aged 19 or above were included; what was the rationale for including, most probably, skeletal mature individuals?

3. Was there an attempt to correlate RAM to menarchal status in females? This would be an interesting analysis of clinical relevance.

4. Although interesting findings, what do the authors believe clinicians can do with this information? Can information about RAM be of clinical benefit for patients? Or can we rely on assessing Risser, Sanders, PHOS, menarche status etc? Although spinal maturation differs on different levels; do these findings have any additional benefit when estimating prognosis for scoliosis patients?

5. The finding that maturation in fewer vertebra have started in growth spurting girls is interesting, would the authors believe that this affects the rotational stability of the spine? The design of the study limits such conclusions naturally.

6. A table including descriptive statistics for the subjects would be beneficial for readers.

Author Response

This is an interesting paper nicely showing the processes of spinal maturity. The authors have done an excellent job shedding light on this important topic, I do however have some comments and questions.

  1. Were the reasons and conditions for CT-scanning for the subjects of such nature that it could disrupt the skeletal maturity process? In other words, are these subjects representable for healthy adolescents?

Author’s response: Dear reviewer, thank you for your useful comments. Patients were screened for pulmonary, gastro-enteric and trauma (most cases) reasons. If any underlying disease that might have influenced the normal development of the child was found in the medical history, the subject was excluded.

Changes in manuscript: Please refer to line 84-94 in which we describe the in- and exclusion criteria.

  1. A significant number of individuals aged 19 or above were included; what was the rationale for including, most probably, skeletal mature individuals?

Author’s response: We understand that the inclusion of young adults could give the impression that a large group of skeletally mature patients were included. In the literature, however, it has been described that the skeletal maturation of the spine and the ring apophysis may last up to 21 years (Uys et al. 2019 and Woo et al. 2018). Specifically, Woo et al., noticed that some unfused rings at the age of 19 and 20. Therefore, in order to capture the end of skeletal maturation within the population, we also included young adults.

Changes in manuscript:  Please refer to line 89-90.

  1. Was there an attempt to correlate RAM to menarchal status in females? This would be an interesting analysis of clinical relevance.

Author’s response: It is interesting to correlate RAM with menarchal status. This was a retrospective study on radiographic data, menarchal status was not available for this population. Therefore, we were not able to correlate RAM with menarche.

Changes in manuscript:  none.

  1. Although interesting findings, what do the authors believe clinicians can do with this information? Can information about RAM be of clinical benefit for patients? Or can we rely on assessing Risser, Sanders, PHOS, menarche status etc? Although spinal maturation differs on different levels; do these findings have any additional benefit when estimating prognosis for scoliosis patients?

Author’s response: Thank you for your question. As there were no studies that focused on the ring apophysis maturation in different spinal regions, we attempted to describe these maturation patterns on high resolution multiplanar CT images. As CTs cannot be used in the clinical practice for skeletal maturation assessment due to ionizing radiation, a next step is to relate the ring apophysis maturation to other maturity markers and to assess the opportunities to visualize the ring apophysis without radiation (for example with advanced MRI techniques such as synthetic bone MRI (Florkow et al. 2019 and Edmondston et al. 2000). If a radiation-free, accurate imaging method of the ring apophysis becomes available for the assessment of spinal maturation, we believe that this may hold more accurate prognostic value for scoliosis progression compared to the conventional (indirect) proxys for (skeletal) maturation.

Changes in manuscript:  The potential future opportunities are described in the discussion section. Please refer to line 273-277.

  1. The finding that maturation in fewer vertebra have started in growth spurting girls is interesting, would the authors believe that this affects the rotational stability of the spine? The design of the study limits such conclusions naturally.

Author’s response: This is an interesting question that we would like to answer in future studies. We believe this influences the stability of the spine as ring apophysis is already known to be a weak point of the spine. Some studies focused on lumbar ring’s fracture (please refer to Kadam et al. 2017 and Xue-yuan et al. 2011) showing weaknesses in adolescent rings and, as consequence, in adolescent spines. Furthermore, Makino et al. in 2016 focused on ossification in the epiphyseal rings of patients with AIS concluding that there is an asymmetrical ossification, mostly evident in the apical region. As you mentioned, the nature of this study does not allow us to claim this, but future studies will possibly focus on the 3D stability of adolescent spine in correlation with ring apophysis maturation.

Changes in manuscript:  Please refer to line 261-265.

  1. A table including descriptive statistics for the subjects would be beneficial for readers.

Author’s response: Thank you for the contributive comment.

Changes in manuscript:  A table showing descriptive statistic of the subject was added. Please refer to line 150-154.

Reviewer 2 Report

Manuscript ID: jcm-1310557

The paper provides knowledge about the maturation pattern of the spine that can help to understand and manage developmental spine problems. I consider that the current study is of general interest for the readers of the journal, but I have some concerns about the manuscript to be addressed by the authors:

-In material and methods section, study population:  I would appreciate if a description of the cohort is included in a table (age, number/percentage of male/female, relevant information that was considered for the article). Some information is shown in the results section but it should be fully described in the material and methods section.

- Line 161: Authors state that no relevant difference between males and females could be detected from the age of 17, but these results are not in agreement with the results presented in Figure 4. At age 19, 60% of females display stage 4 of maturation and more than 90% of males display the same stage of maturation.

-Line 193:Spearman’s test instead of Spearmen’s test.

-Line 223: Please add the complete word for AIS as it has not been previously defined in the manuscript.

-Discussion section: The authors state that ‘The area of the spine in which most common types of idiopathic scoliosis develop, appears to mature later than the rest of the spine, which may be important for understanding the patho-mechanism of idiopathic scoliosis’. I would appreciate if the authors could discuss in more detail this point and explain which are the possible mechanisms for the authors relating scoliosis and maturation of the spine.

Author Response

The paper provides knowledge about the maturation pattern of the spine that can help to understand and manage developmental spine problems. I consider that the current study is of general interest for the readers of the journal, but I have some concerns about the manuscript to be addressed by the authors:

-In material and methods section, study population:  I would appreciate if a description of the cohort is included in a table (age, number/percentage of male/female, relevant information that was considered for the article). Some information is shown in the results section but it should be fully described in the material and methods section.

Author’s response: As suggested by both reviewers, a table showing demographics and descriptive statistics was added. As we structured the manuscript in a way in which descriptive statistic are shown in the results and not in the methods (based on journal’s guidelines), we prefer to keep it this way. For better clarification we included more information about exclusion criteria and the age range in our methods.

Changes in manuscript:  Please refer to line 150-154 for tablePlease refer to line 89-90 and 93.

- Line 161: Authors state that no relevant difference between males and females could be detected from the age of 17, but these results are not in agreement with the results presented in Figure 4. At age 19, 60% of females display stage 4 of maturation and more than 90% of males display the same stage of maturation.

Author’s response: We would like to make our sincere apologies. The sentence was updated with the correct information.

Changes in manuscript:  Please refer to line 170-171.

-Line 193:Spearman’s test instead of Spearmen’s test.

Changes in manuscript:  Please refer to line 202.

-Line 223: Please add the complete word for AIS as it has not been previously defined in the manuscript.

Changes in manuscript:  Please refer to line 232.

-Discussion section: The authors state that ‘The area of the spine in which most common types of idiopathic scoliosis develop, appears to mature later than the rest of the spine, which may be important for understanding the patho-mechanism of idiopathic scoliosis’. I would appreciate if the authors could discuss in more detail this point and explain which are the possible mechanisms for the authors relating scoliosis and maturation of the spine.

Author’s response: Thank you for your nice comment. Sharpey’s fibers insert into the ring apophysis and anchor the intervertebral disc to the two adjacent vertebrae. We believe this is an important weak point that has important implications for the mechanical stability of the disc-vertebral body complex of the mid-thoracic and thoraco-lumbar sections at a time that spinal loading increases rapidly due to growth spurt.

Changes in manuscript: Please refer to line 261-265.